# A Cluster Analysis of Forward Trajectory to Identify the Transport Pathway of Salt-Dust Particles from Dried Bottom of Aral Sea, Central Asia

**Aishajiang Aili** , **Jilili Abuduwaili** , **Hailiang Xu \***, **Xinfeng Zhao and Xinghong Liu**

State Key Laboratory of Desert and Oasis Ecology, Xinjiang Institute of Ecology and Geography,
Chinese Academy of Sciences, Urumchi 830011, China; aishajiang@ms.xjb.ac.cn (A.A.); jilil@ms.xjb.ac.cn (J.A.);
zhaoxf@ms.xjb.ac.cn (X.Z.); lixh@ms.xjb.ac.cn (X.L.)
**\*** Correspondence: xuhl@ms.xjb.ac.cn; Tel.: +86-991-782-7350; Fax: +86-991-788-5320

**Abstract:** The Aral Sea basin is the most active source of salt-dust storms in the central Asian region, while its exposed bottom is acting as a "distributer" of salts and chemicals over the adjoining areas. In this study, the Hybrid Single Particle Lagrangian Integrated Trajectory Model (HYSPLIT-4) is used to identify the trajectories of air parcels from the dried bottom of the Aral Sea region (45.40° N, 61.30° E) that are potentially containing salt-dust and their probability of influencing the downwind area in the period of 2016–2020. The frequency of air parcel trajectory was mapped for six levels: 100, 300, 500, 1000, 2000, and 3000 m agl. The trajectories were categorized by *k-means* clustering into four clusters that are named by their direction of movement as follows: Cluster 1: E category, Cluster 2: NE category, Cluster 3: W category, and the Cluster 4: S category. The 72 h of forward trajectories showed that salt-dust storms starting from the dried bottom of the Aral Sea had the highest probability of affecting the northeastern region e.g., Siberian Plain, followed by the southern region e.g., Iran Plateau. Total number of trajectories within these two clusters (NE and S) accounts for 90% (or 413 days) of trajectories in examined days. The main area of influence of salt-dust is close to the source area. The potential transport distance of salt-dust particles increases with the height of the starting point. The surface wind, which results from the changes of the Siberian High (SH), has a major role in shaping the surface atmospheric circulation which determines the transport pathway of salt-dust particles over the Aral Sea region. The results of this study could be useful to forecast the potential occurrence of salt-dust storms in downwind affected areas and would also be helpful to understand the possible causes of salt-dust storms which can provide the scientific basis for mitigation of the negative impact of salt-dust storms on the environment and human health. Further research should be conducted by using monitoring data to confirm the deposition of dust and salt particles in those areas mapped by our study.

**Keywords:** cluster analysis; forward trajectory; salt-dust particles; transport pathway; Aral Sea

## 1. Introduction

The central Asian region is highly affected by salt-dust storms. The ecological disaster occurring at the Aral Sea region, the fourth largest lake in the world, has been realized since 1990s N [1,2]. The extreme drought conditions with very low precipitation (100–250 mm·a$^{-1}$) in the Aral Sea region exacerbated the desertification process which resulted in the formation of a new desert called the Aralkum in the dried bottom of the sea [3]. The dried bottom of the Aral Sea has emerged as a new "hot spot" of dust and salt storms in this region. Each year, huge amounts of dust and salt particles disperse from the bottom of the Aral Sea into the atmosphere or travel with the wind to far regions [3–5]. At present, the dried bottom of the Aral Sea has become a major source of dispersing the salts and other chemicals to the adjoining areas. Resultant salt storms, known as "white dust storms" [6], are now polluting the atmosphere, water, and soil which in turn deteriorating the natural

ecosystem. However, the wind condition, which is mainly determined by the Siberian High (SH) pressure system in the Aral Sea region, plays an essential role in determining the movement direction of salt-dust storms. The Siberian High is an anticyclone generated in the lower troposphere over the Siberian-Mongolian area (42.5° N–57.5° N, 80° E–120° E) during winter. It is an important factor of the Eurasian climatic system, which interacts in central Asia [7–9]. Variations of SH not only have impact on the hydrological cycle but also determine the dust entrainment, transport, and deposition on the Eurasian continent. In general, the SH pressure system begins to build up at the end of August, reaches peak in the winter, and remains strong until the end of April [10–12]. This synoptic situation induces strong wind, hence higher frequency of salt-dust storms in the study area. There are several methods to detect the transport pattern of air parcels [13–15]. Among various research tools, the HYSPLIT model is the most widely used air trajectory model which can trace the source–receptor relationship over long distances. This model has the ability to compute simple air parcel trajectory to identify the complex dispersion and deposition processes.

The HYSPLIT model was initially developed by the joint efforts of the National Oceanic and Atmospheric Administration (NOAA) of USA and Australia's Bureau of Meteorology. The HYSPLIT model (http://www.arl.noaa.gov/HYSPLIT_info.php (accessed on 24 December 1997).) can be used to find the advection or trajectory of a single pollutant particle without any additional dispersion module [16–18]. In previous studies on the dust emission and extension from the dried bottom of Aral Sea, different techniques like surface particulate monitoring, satellite data processing, elemental tracer's analysis, and dust dispersion modeling have been widely used [19–21]. Most of these studies have concentrated on finding the origins of the dust, and investigating the mowing pathway of salt particles [22]. However, limited research has been conducted to identify the long-range transport pathway of the air parcel from the Aral Sea region. This study therefore conducted to partly fill in this information gap. The HYSPLIT forward trajectory model is used to identify the direction and extension of the air parcel from the dried bottom of the Aral Sea, and the trajectory pathways of air parcels were analyzed combined with the atmospheric pressure system.

## 2. Materials and Methods

### 2.1. Description of Aral Sea

The Aral Sea is situated between the two countries Kazakhstan and Uzbekistan in central Asia (Figure 1). This region has a continental dry climate because it is located far away from the ocean. Summer temperature reaches 40 °C degrees while the temperature in winter falls below −20 °C. The mean annual precipitation is 100–250 mm, the average monthly precipitation ranges from 6 mm (in September) to 15 mm (in March). The main source of surface waters is thaw water coming from high glaciers which also feeds the Amu River and Syr River, the two largest rivers in the region [23]. Human impacts on water quantity and quality are apparent. Regarding agricultural activity, a large amount of water has been withdrawn for irrigation from the Amu River and Syr River that subsequently flows back into the rivers. Moreover, the arid climate conditions exacerbated the desertification process which resulted in the formation of a new desert called the Aralkum in the dried bottom of Aral Sea. In the 1960s, the Aral Sea having an area of 68,000 km$^2$ was the 4th biggest lake in the world. However, lack of recharge from rivers and enormously high evaporation in summer resulted in steady shrinkage of Aral Sea after the 1960s. Till 2007, the Aral Sea water area decreased to only 10% of its original size and fragmented into four lakes, namely the eastern and western basins of the South Aral Sea, the North Aral Sea, and a small lake between the North and South Aral Sea [24]. By 2009, the southeastern lake vanished and the southwestern lake turned into a thin strip at the extreme west of the southern sea. It is also observed that in subsequent years, the southeastern lake has been occasionally replenished with water to a small degree. In August 2014, for the first time, NASA released the satellite images indicating that the eastern basin of the Aral Sea had become completely dry and this basin is now called the Aralkum Desert.

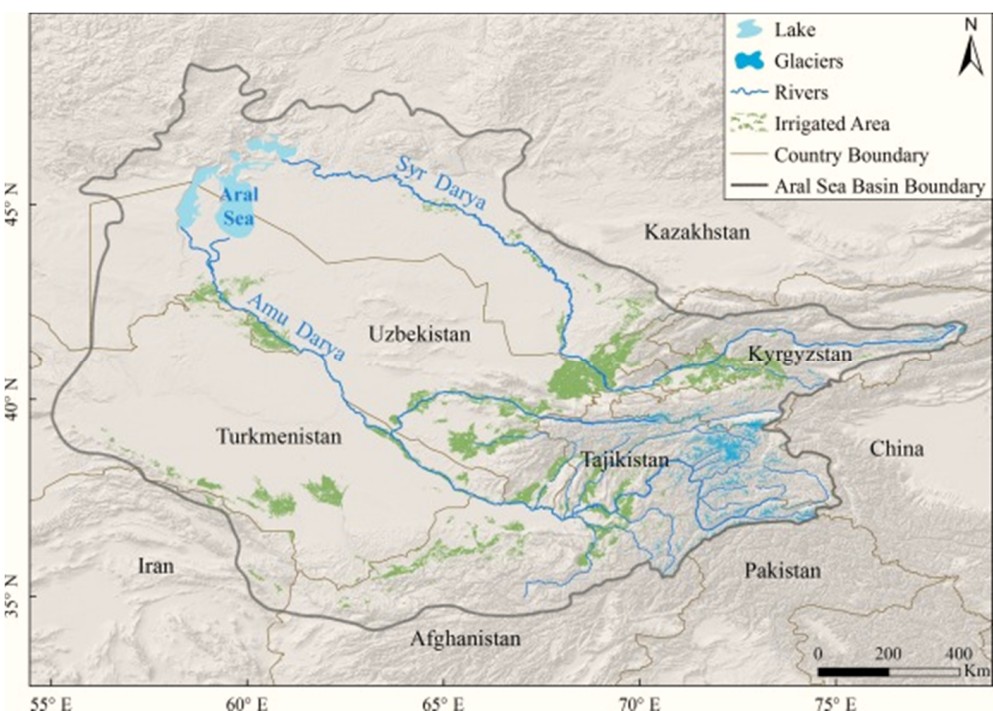

**Figure 1.** Location of the Aral Sea.

### 2.2. Data and Method

Data used in this study were extracted from the National Oceanic and Atmospheric Administration, Air Resources Laboratory (NOAA ARL), which are available at https://www.ready.noaa.gov/hypub-bin/trajtype.pl (accessed on 30 September 2020). The 72 h forward trajectories were obtained using HYSPLIT4, available online at http://www.arl.noaa.gov/ready/hysplit4.html (accessed on 24 December 1997) for a total of 460 days from 2016 to 2020. Three months (from 1 March to 31 May, total of 92 days) were selected from each year because of high salt storm occurrence in these month. Thus, a total of 460 days were selected for the HYSPLIT model analysis. The northeast dried bottom of the Aral Sea was selected as the starting point due to this area having been identified as a salt-dust storm source region in central Asia [3,25–30]. The trajectory starts from latitude of 45.4° N and longitude of 61.3° E at 9:00 UTC (Coordinated Universal Time) each day, which is equivalent to 14:00 p.m. local standard time (LST). To minimize the friction effect from the Earth's surface and to represent wind in the lower boundary layer, the starting level of air masses is fixed at 100 m above ground level (AGL). The "Final Run" meteorological data archives (FNL) of the Air Resource Laboratory, National Oceanic and Atmospheric Administration (NOAA), for the northern hemisphere, which are available online at ftp://www.arl.noaa.gov/pub/archives/fnl/ (accessed on 24 December 1997), were used to run the HYSPLIT4 model.

The classification of trajectories of air mass was performed using the *k*-means clustering technique (SPSS 21) on a dataset consisting of 6 meteorological variables, including ambient temperature, potential temperature, daily rainfall, mixing layer depth, relative humidity, and solar radiation flux at different latitude, longitude, and altitude of the trajectory. The coordinates of air mass forward trajectories used in this study include latitudes, longitudes, and altitude of the air masses at 6 points on a trajectory: 12, 24, 36, 48, 60, and 72 h. This set of 18 trajectory coordinates was selected from several trials because it satisfactorily reproduced the spatial positions of the air mass trajectories. Thus, the final dataset used for classification consists of 460 days × (6 + 18) variables. The Principal Component Analysis (PCA) technique with Varimax rotation was applied [15–17,31] on the normalized data set.

For example, let $X = [x_i]$ be any $k \times 1$ random vector. We now define a $k \times 1$ vector $Y = [y_i]$, where for each $i$, the $i$th principal component of $X$ is:

$$y_j = \sum_{j=1}^{k} \beta_{ij} x_j \tag{1}$$

where $\beta_{ij}$ is a regression coefficient; since each $y_i$ is a linear combination of the $x_j$, $Y$ is a random vector.

The $k$-means clustering technique following the score matrix of components produced through PCA was employed to identify the pattern of air mass trajectories. The main objective of $k$-means clustering is to split the dataset into a certain number of homogeneous clusters ($k$). The cluster membership can be determined based on the distance between the data point and the $k$th centroids.

In the clustering process, the spatial variance (SV) is computed between each endpoint (k) along the trajectory (j) within its cluster (i).

$$SV_{i,j} = \sum_k (P_{j,k} - M_{i,k})^2 \tag{2}$$

where the P and M are the position vectors for the individual trajectory and its cluster mean trajectory.

## 3. Results and Discussions

### 3.1. Classification of Air Parcel from Dried Bottom of Aral Sea

The HYSPLIT forward trajectories were obtained for every 460 days, and examples are provided in Figure 2.

Based on the score matrix of the 72 h of air mass transport pathway obtained by using the HYSPLIT model, the k-means clustering technique classified the air mass trajectories starting from the northeast dried bottom of the Aral Sea at 460 days of the study period (2016–2020) into 4 clusters. These 4 clusters are quite different in terms of shape, length, vertical position of the 72 h forward trajectory, as shown in Figures 2 and 3. However, a satisfactory uniformity regarding shape, origin, pathway, and trajectory length has been noted in each cluster. Based on the movement direction and pathway of each cluster of air mass, which were obtained by using PCA and the k-means clustering technique, the trajectory pattern of air mass was classified into four categories: Cluster 1: (E category), Cluster 2: (NE category), Cluster 3: (W category), and Cluster 4: (S category).

### 3.2. Occurrence Frequency and Meteorological Characteristics of Each Cluster

The variations of 72 h of forward trajectories obtained for a particular month are quite typical and do not change much from year to year during the 5 years period. The monthly occurrence frequency and meteorological conditions of each cluster are presented in Table 1.

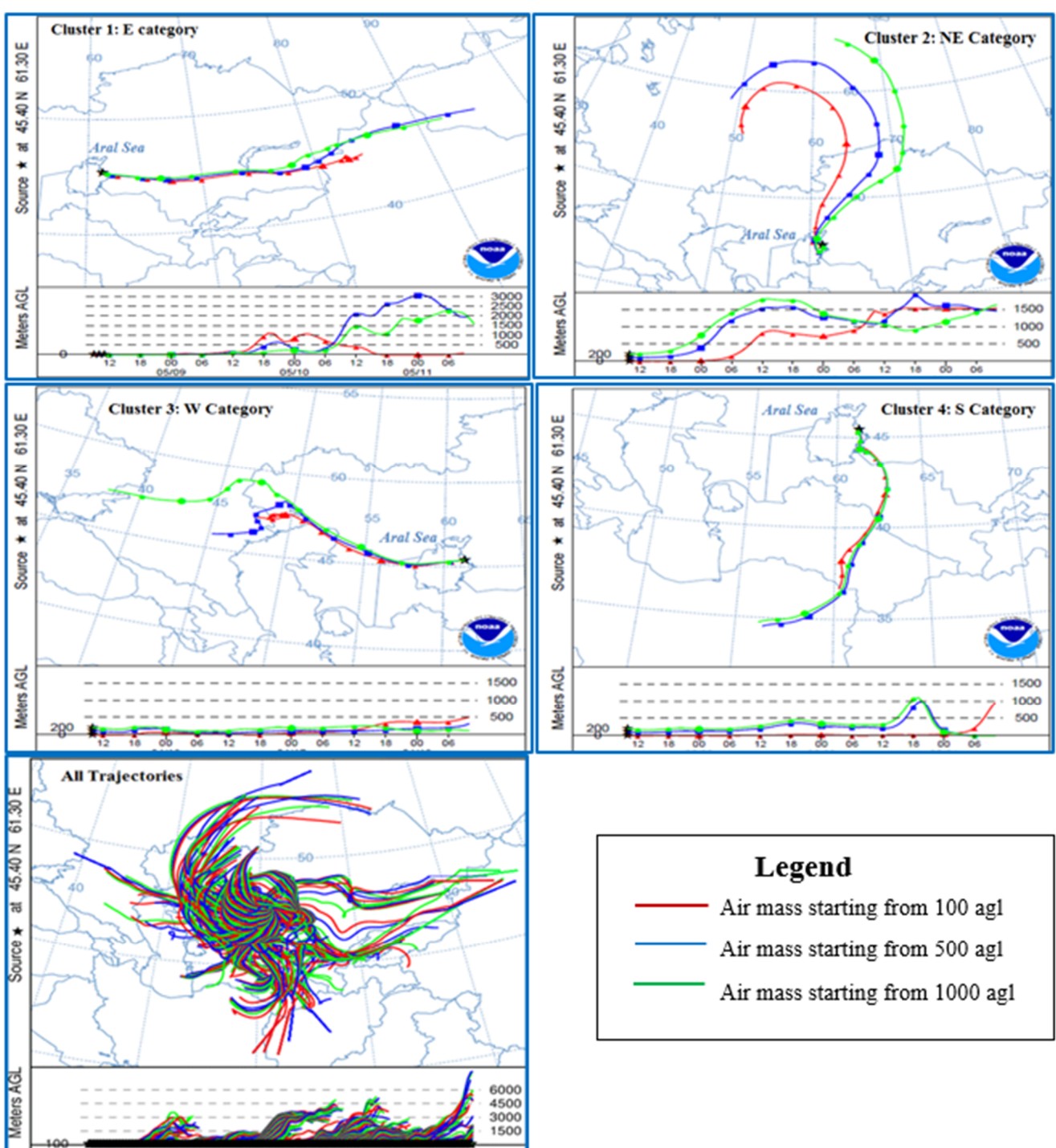

**Figure 2.** Trajectory sample for each cluster.

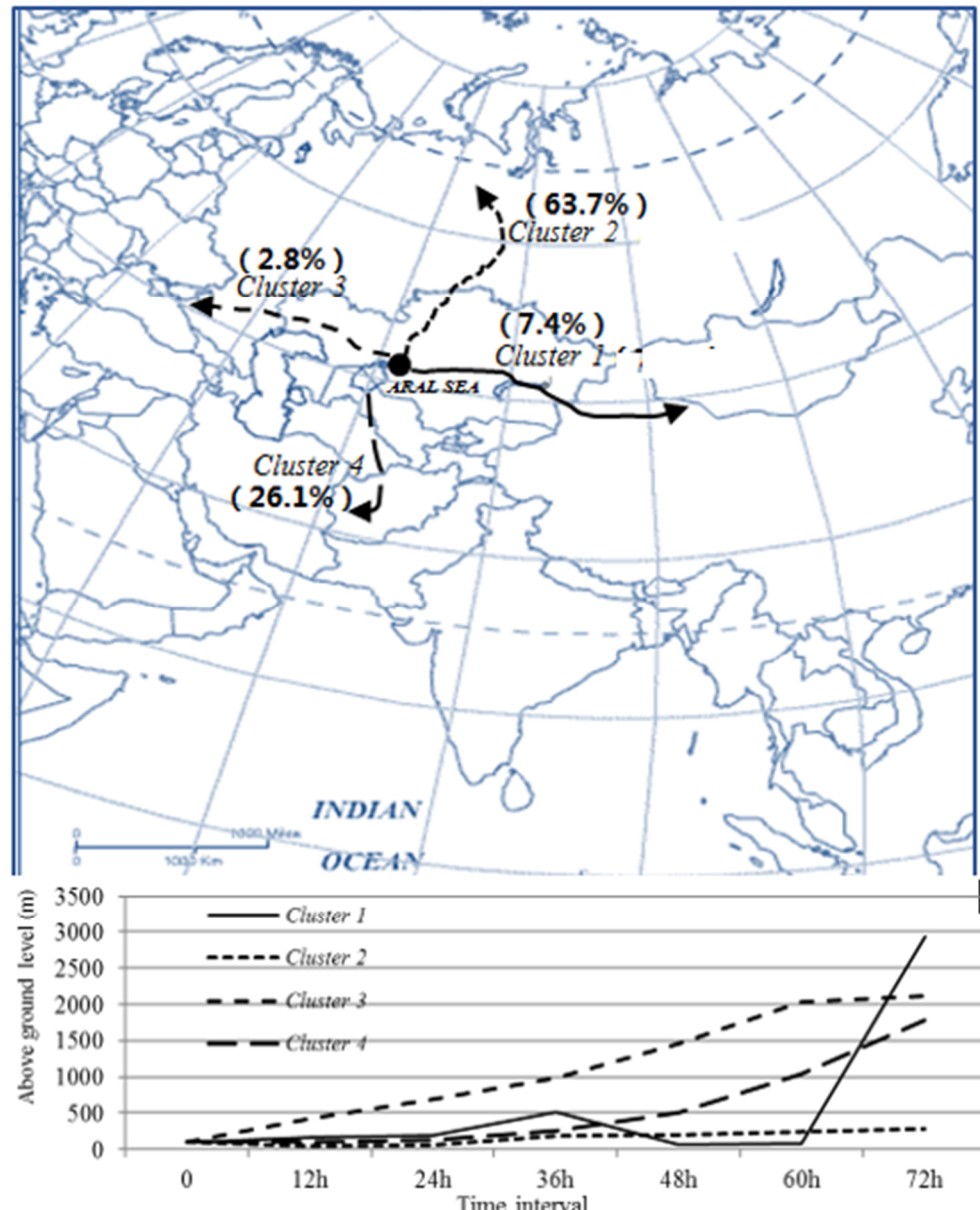

**Figure 3.** Trajectory pattern of each cluster and their altitudes along with pathway.

**Table 1.** Occurrence frequency and meteorological characteristics of each cluster.

| Variables | Clusters | | | |
|---|---|---|---|---|
| | E Category | NE Category | W Category | S Category |
| Number of observed days | 34 | 293 | 13 | 120 |
| March | 7 | 156 | 3 | 19 |
| April | 9 | 102 | 8 | 34 |
| May | 18 | 35 | 2 | 69 |
| Potential temperature (K) | 292.68 | 289.02 | 287.10 | 292.76 |
| Ambient temperature (K) | 293.40 | 290.95 | 287.77 | 293.47 |
| Rainfall (mm/h) | 0.00 | 0.02 | 0.00 | 0.00 |
| Mixing layer depth (m) | 815.19 | 832.08 | 697.47 | 918.16 |
| Relative humidity (%) | 37.88 | 43.60 | 43.17 | 41.47 |
| Solar radiation flux (W/m$^2$) | 796.68 | 733.07 | 768.73 | 734.71 |

The transport pathway and meteorological characteristics of each cluster are as follows:

(1)    Cluster 1: E category.

Air masses of this category starting from the northeast part of the dried bottom of the Aral Sea, and move eastward through Kazakhstan territory, then turn east and enter China territory through the lower altitude of mountain areas, and then continuously move eastward along the north slope of Tianshan Mountain, arrive at Ili valley, Jungar Basin of Xinjiang, China and west part of Mongolia with highest altitude (2600 m).

This cluster was observed on 34 days out of 460 total dusty days (around 7.4%), mainly observed in the period of May when the influence of the NE cold air mass is more significant. It can be characterized by the dry and hot air masses with the high potential temperature (292.8 K) and ambient temperature (293.4 K) as well as the lower rainfall (0 mm/day) and lowest relative humidity (37.88%) (Table 1).

(2)    Cluster 2: NE category.

This air mass starting from the northeast part of the dried bottom of the Aral Sea moves to the northeast and north direction along the west slope of Ural Mountain with lower altitudes (70–250 m), arrives at the middle part of the west Siberian Plain of Russia (Figures 2 and 3). This cluster occurs with highest frequency (293 days) of all four clusters and accounts for 63.6% of the entire trajectory in the examined days. The air masses of this cluster can be characterized by humid air mass with the highest rainfall (0.02 mm/h) and highest relative humidity (43.60%) (Table 1).

(3)    Cluster 3: W category.

Air masses in this cluster move westward with the lowest altitude among all of the clusters, pass over lowland in the north shore of the Caspian Sea, and finally arrive at the west shore of the Black Sea (Figures 2 and 3).

This cluster occurs with the lowest frequency of all clusters (13 days or 2.8%) and is predominantly observed in the period of April. This air mass type is observed with the lowest potential temperature (287.10 K) and ambient temperature (287.77 K) as well as the lowest mixing layer depth (697.47 m) (Table 1).

(4)    Cluster 4: S category.

Air masses within this cluster move southward with lower altitude, pass over the Turan lowland area, and then turn southwest with higher altitude and arrive at the middle part of the Iran Plateau (Figures 2 and 3).

This cluster type was observed with the second highest frequency on 120 days out of 460 total examined days (accounts for 26.1%). It is observed mainly in the period of May, and can also be characterized by hot air masses with a highest potential temperature (292.47 K) ambient temperature (293.76 K) and highest mixing layer depth (918.16 m) (Equation (1)). Our results are consistent with other studies. Karami et al. analyzed two characteristic dust-storm events originated from these areas on 12–15 July 2016 and 27–28 May 2018 by using the DREAM-NMME-MACC, CAMS, and WRF-Chem model, and found that, during the Aralkum dust storm on 27 May, HYSPLIT outputs indicated the prevailing winds were from northern directions, so dust plumes highly affected the southern parts of central Asia and entered eastern Iran [24].

It can also be seen from Figure 4 that the main area of influence of salt-dust is close to the source area, and the higher the starting point, the longer the transport distance. Regardless of the starting height of trajectories, the trajectory frequency shows higher value, e.g., >90%, in closer areas of the Aral Sea. The transport distance of the air parcel is increased with the height of the starting point. When the starting point is 100 m agl, the coverage areas of trajectory are between the 35° N–50° N and 45° E–90° E, while when the starting point is 2000 and 3000 m agl, the coverage areas of trajectory extended to vast areas, e.g., 30° N–75° N and 60° E–115° E.

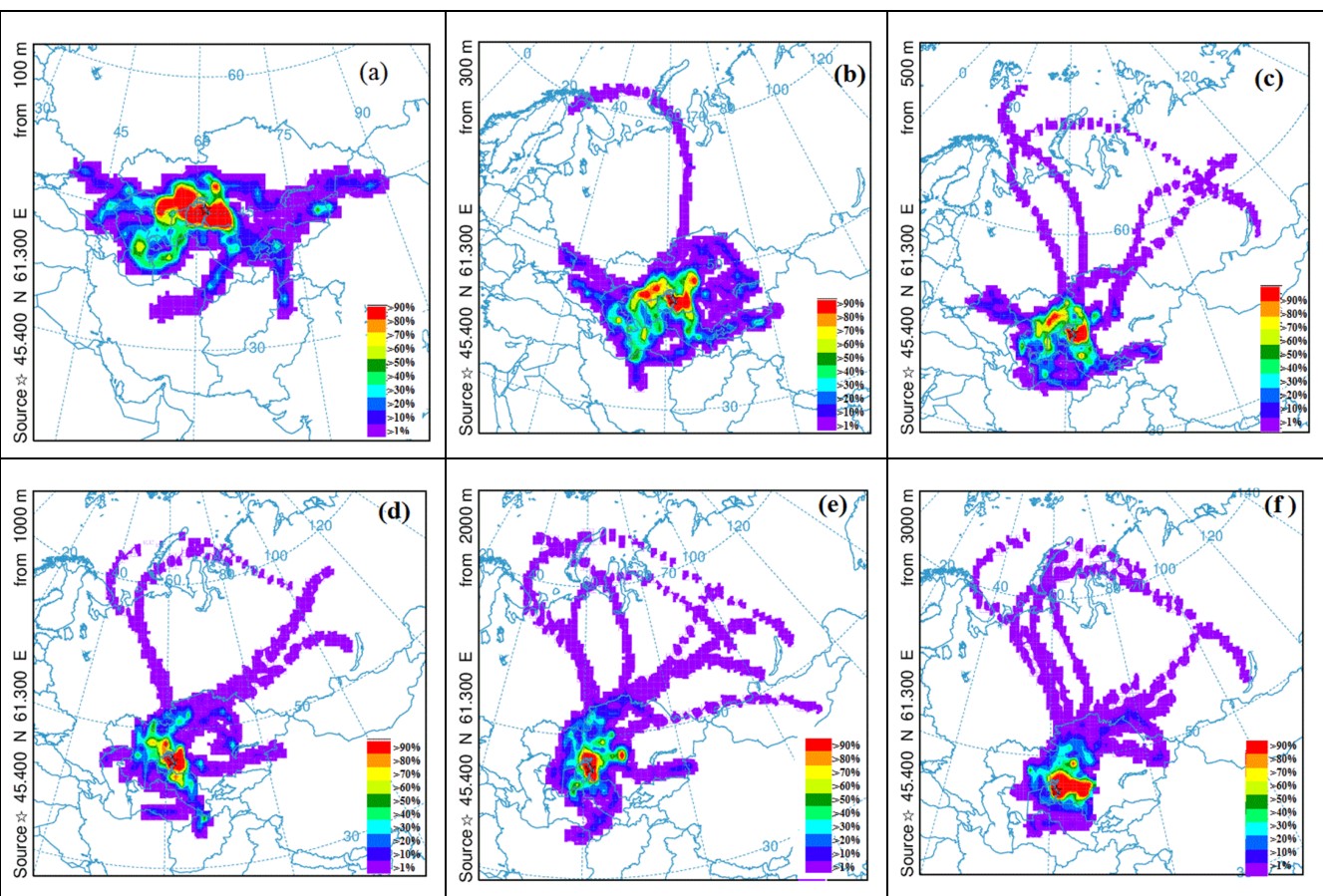

**Figure 4.** Occurrence frequency of 72 h of forward trajectory of air masses starting from different AGL (above ground level) of dried bottom of the Aral Sea: (**a**) 100 m agl.; (**b**) 300 m agl.; (**c**) 500 m agl.; (**d**) 1000 m agl.; (**e**) 2000 m agl.; (**f**) 3000 m agl.

### 3.3. Relationships between Air Mass Clusters and Synoptic Pressure System

The synoptic pressure patterns of each trajectory during the study period were also analyzed. This difference in temperature results in a difference in pressure which initiates the movement of sand and salt particles from the dried bottom of the Aral Sea to downwind areas. To examine the atmospheric circulation over the Aral Sea region, we extracted the Asian pressure maps corresponding to each clusters using NCEP/NCAR (National Center for Atmospheric Research/National Center for Environmental Prediction, Boulder, CO, USA) data, reanalyzed the trajectory pattern of each cluster within different air pressure conditions. It is evident that the prevailing surface winds over the Aral Sea are northeaster in March and early April and southwestern at the end of April, May, and June. This change is related to the Siberian High (SH), which is the most dominant Northern Hemisphere anticyclone and is centered between 40 and 65° N, 80 and 120° E. It plays a key role in shaping surface atmospheric circulation over the Eurasian continent. In spring, when the SH retreats and decreases, the main surface winds over the Aral Sea turn to the southwest (Cluster 3, Figure 5). During the whole study period, the pressure system of air between the Aral Sea region and surrounding area is quite different. Examples of the synoptic pattern for each trajectory are shown in Figure 5.

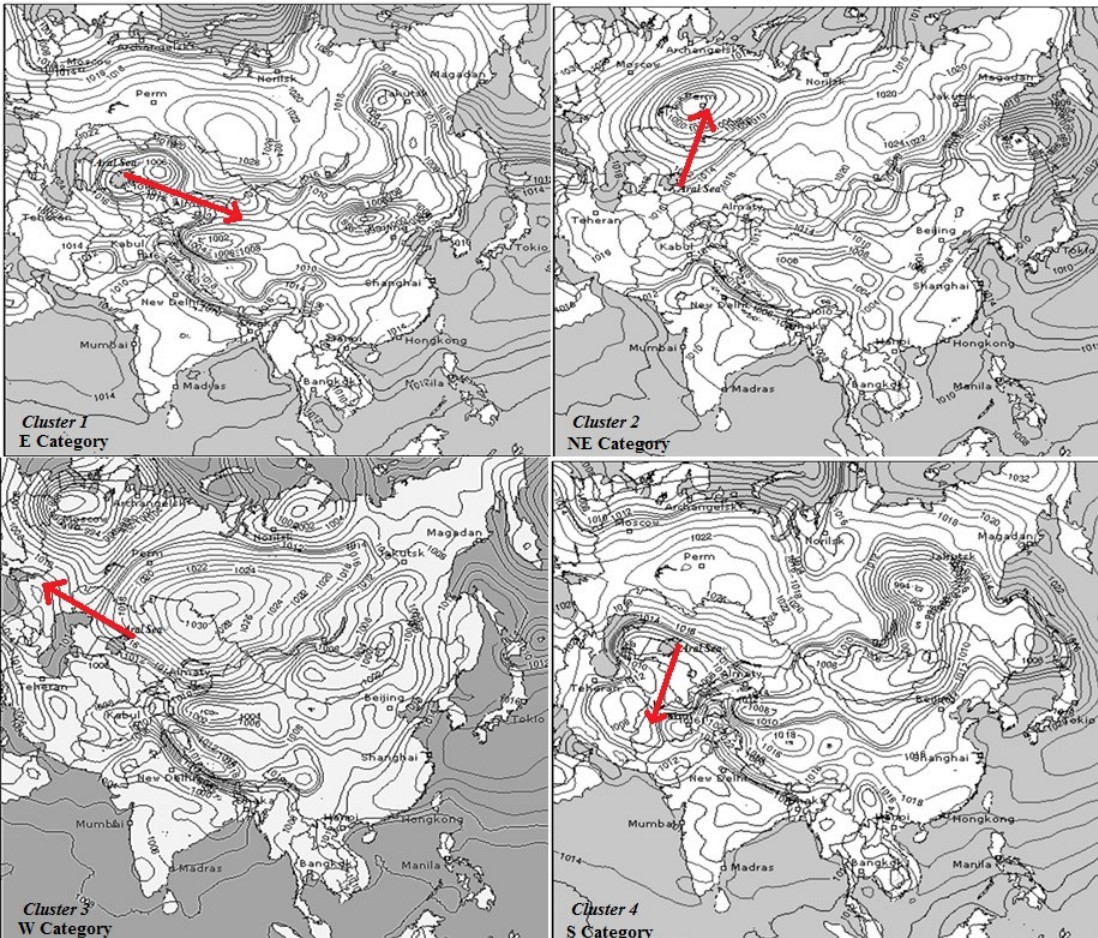

**Figure 5.** Examples of synoptic pressure pattern for each cluster. Note: the red arrow in the figure indicates the movement direction of air parcel within different air pressure system.

It can be seen from Figure 5 that, the pressure systems of air between the Aral Sea region and surrounding area in different months are quite different. Clusters 1 and 4 are mostly observed in the month of May when the influence of NW moist air mass from the Atlantic Ocean is increased. In this month, trajectories of this cluster are stretched towards the east and south direction. The synoptic map of these clusters (Figure 5) indicates a strong influence of high and low pressure areas with dense isobars. During this period, the dried bottom of the Aral Sea was under the influence of a pressure ridge, extending from a high pressure system ($p$ = 1022 mb), while downwind areas belonging to these clusters experienced a low pressure system (1004–1010 mb) during the same period. In contrast to these two clusters, Cluster 2, which was mainly concentrated in the months of March and April when the influence of warm air masses from the Indian Ocean are increased, has a typical synoptic map that shows the influence of a vast low pressure system in the north (1008 mb) and high pressure system in the south (1018 mb) with dense isobars (Figure 5). Similar results are proposed by other studies. For example, Karami et al. analyzed the relationships between the air pressure system and salt-storm movement from the dried bottom of the Aral Sea by using the WRF-Chem and ERA-Interim model, and found that, the Siberian high trough dominated at the northern latitudes, while a ridge with warm air was developed over southern Iran and the Arabian Sea. This combination resulted in an enhanced pressure gradient over arid central Asia, with strengthening the zonal wind over the Aralkum Desert, which may transfer dust aerosols over the mountainous ranges of Pamir and TianShan or even over China [24].

In this study, the potential impact region represented by the transport pathway is quite vast. Therefore, further research needs to be conducted to explain whether salt-dust particles can travel to these areas and how much of these particles then settle there.

## 4. Conclusions

Using the HYSPLIT forward trajectory model, we have analyzed the movement and frequency of air parcels originating from the dried bottom of the Aral Sea for a 5-year period and its affected downwind regions within 72 h. The results of this study indicated that air parcel trajectories originating from the dried bottom of the Aral Sea can affect regions many thousands of kilometers in a relatively short period of time. Moreover, under favorable meteorological conditions, these air parcels have the potential to transport dust and other aerosols to these areas. Forward trajectories showed that dust storms starting from the dried bottom of the Aral Sea had the highest probability of affecting the northeastern region e.g., Siberian Plain, followed by the southern region e.g., Iran Plateau. The total number of trajectories within these two clusters (NE and S) accounts for 90% (or 413 days) of trajectories in examined days. Although small numbers of trajectories were observed within E and W categories, the air masses within these two clusters had a potential to transport the salt-dust particles from the dried bottom of the Aral Sea to the Jungar Basin in western China and the Caspian Sea region. The main area of influence of salt-dust is close to the source area, and the higher the starting point, the longer the transport distance of air parcel trajectory. Further research needs to be conducted by using monitoring data to confirm the deposition of dust and salt particles in the areas mapped by our study.

**Author Contributions:** Conceptualization, A.A.; Data curation, H.X. and J.A.; Formal analysis, X.Z. and X.L.; Methodology, A.A. and H.X.; Project administration, A.A. and H.X. All authors have read and agreed to the published version of the manuscript.

**Funding:** This research was financially supported by the Western Light Foundation of Chinese Academy of Sciences (Y734341).

**Institutional Review Board Statement:** Not applicable.

**Informed Consent Statement:** Not applicable.

**Data Availability Statement:** The data that support the findings of this study are available from the corresponding author, upon reasonable request.

**Acknowledgments:** This study is financed by the Western Light Foundation of Chinese Academy of Sciences (Y734341). The authors would like to acknowledge the NOAA air resource laboratory for providing HYSPLIT online.

**Conflicts of Interest:** The authors declare no conflict of interest.

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
