# Peer review of "A Cluster Analysis of Forward Trajectory to Identify the Transport Pathway of Salt-Dust Particles from Dried Bottom of Aral Sea, Central Asia"

_atmosphere, doi:10.3390/atmos12060764_

Round 1

Reviewer 1 Report

I suggest to include in the Introduction a brief description of the anual dynamics of SH, given that this important pressure system determines the meteorology of the regions. Then, you can introduce your study in relation to the spring position of the SH, and its changes from early to latae spring. This is an important result of this study and should be highlighted.

Author Response

Thank you very much for your valuable comment. The manuscript is revised as your comment, and highlighted with green color.

The Siberian high (SH) is an anticyclone generated in the lower troposphere over the Siberian-Mongolian area (42.5°N–57.5°N, 80°E–120°E) during winter. It is an important factor of the Eurasian climatic system, which interact in the central Asia. Variations of SH not only have impact on the hydrological cycle but also determine the dust entrainment, transport and deposition on the Eurasian continent. In general, the SH pressure system begins to build up at the end of August, reaches peak in the winter, and remains strong until the end of April. This synoptic situation induces strong wind hence higher frequency of salt-dust storm in study area. This explanation and related references are added in revised manuscript, accordingly (line 45 – 55 and reference 3,7,8,9 at revised manuscript).

Reviewer 2 Report

There are no line numbers, which will make it difficult for authors to locate these comments. This is an interesting paper but much more is needed on the background situation of the Aral Sea and in relating the findings back to the literature.

  1. Introduction

The ecological disaster occurring at the Aral Sea region, FORMERLY the fourth largest lake in the world, has been realized SINCE 1990s [reference 1 in the list of refs: the first author is spelled Orlovsky, not Oriovsky, AND SOME EARLIER REFENCES NEED TO BE CITED.

‘extreme drought conditions in the Aral Sea’ I’M NOT CONVINCED THAT DROUGHT WAS IMPORTANT – IRRIGATION OFFTAKES WERE MUCH MORE IMPORTANT. AND WHEN WAS THIS DROUGHT?

There are plenty of earlier references that can be cited to give a much better picture of the Aral Sea disaster here and in section 2.1 (e.g. Micklin 1988, Glazovsky 1995, Middleton 2002)

2.1. Description of Aral Sea

WHEN DID THE ARAL SEA HAVE ‘an area of 68,000 km2’?

  1. Results and Discussions

THE RESULTS ARE INTERESTING BUT THE DISCUSSION NEEDS TO REFER BACK TO THE LITERATURE. Ge et al. (2016), Shukurov & Shukurova (2019) and Karami et al. (2021) COME TO MIND BUT THERE ARE PROBABLY MORE.

references

Ge, Y., Abuduwaili, J., Ma, L., & Liu, D. (2016). Temporal variability and potential diffusion characteristics of dust aerosol originating from the Aral Sea basin, central Asia. Water, Air, & Soil Pollution, 227(2), 63.

Glazovsky, N.F. (1995) The Aral Sea basin. In: Kasperson, J.X., Kasperson, R.E. and Turner II, B.L. (eds), Regions at risk: comparisons of threatened environments. Tokyo, United Nations University Press: 92-139.

Karami, S., Hamzeh, N. H., Kaskaoutis, D. G., Rashki, A., Alam, K., & Ranjbar, A. (2021). Numerical simulations of dust storms originated from dried lakes in central and southwest Asia: The case of Aral Sea and Sistan Basin. Aeolian Research, 50, 100679.

Micklin, P.P. (1988) Desiccation of the Aral Sea: a water management disaster in the Soviet Union. Science, 241: 1170-1176.

Middleton. N. (2002) The Aral Sea. In Shahgedanova, M. (ed) The Physical Geography of Northern Eurasia. Oxford University Press: 497-510.

Shukurov, K., & Shukurova, L. (2019). Aral's potential sources of dust for Moscow region. In E3S Web of Conferences (Vol. 99, p. 02015). EDP Sciences.

Author Response

Thank you very much for your valuable comment. The manuscript is revised as your comment and the revised part is highlighted with green color. Detailed responses for each specific comments are attached. 

This manuscript is a resubmission of an earlier submission. The following is a list of the peer review reports and author responses from that submission.

Round 1

Reviewer 1 Report

There are some ambiguities in the description of the results, maybe lapsus, maybe difficulties with the language. I have made many comments in the annotated text that seek to clarify those ambiguities. I also propose to rearrange Figures since Fig 4 is strongly related to Fig 2, while Fig 3 reports on data that come later in the presentation.

I have a question regarding the classification. I understand that clusters are obtained based on the PCA, and 4 clustrers result. But in the text, it is also stated that a "visual" classification is made in categories: E, NE, W and S. How were obtained these categories?

Figures 2,3,5 need better captions, e.g. they should be labelled a), b), c) d)..etc.

Bibliography citations: revise punctuation
